# Evaluation of a Phytogenic Feed Supplement Containing Carvacrol and Limonene on Sheep Performance and Parasitological Status on a Hungarian Milking Sheep Farm

**DOI:** 10.3390/vetsci10060369

**Published:** 2023-05-23

**Authors:** Éva Varga-Visi, Gábor Nagy, Ágnes Csivincsik, Tamás Tóth

**Affiliations:** 1Institute of Physiology and Animal Nutrition, Kaposvár Campus, Hungarian University of Agriculture and Life Sciences, 7400 Kaposvár, Hungary; vargane.visi.eva@uni-mate.hu (É.V.-V.); csivincsik.agnes@uni-mate.hu (Á.C.); 2Agribiotechnology and Precision Breeding for Food Security National Laboratory, Kaposvár Campus, Hungarian University of Agriculture and Life Sciences, 7400 Kaposvár, Hungary; 3One Health Working Group, Kaposvár Campus, Hungarian University of Agriculture and Life Sciences, 7400 Kaposvár, Hungary; 4ADEXGO Ltd., 8230 Balatonfüred, Hungary; tamas.toth@adexgo.hu

**Keywords:** carvacrol, limonene, sheep, performance, *Haemonchus contortus*, plant extract

## Abstract

**Simple Summary:**

The inclusion of plant-derived products, such as herbs or essential oils, in the feed can improve the production of animals. Moreover, these photogenic products can fight against parasites in an animal’s digestive system. The scope of the present study was to observe the impact of a feed supplement containing carvacrol and limonene on the performance and parasitological status of sheep. The feed supplement exerted an advantageous effect on the metabolic processes of lactating ewes after 42 days of supplementation, while the average live weight and average daily weight gain of their twin suckling lambs increased by the termination of the study. In another experiment, fattening lambs fed the same supplement showed a decreased faecal egg count of gastrointestinal nematodes. Additionally, no differences in live weight, average daily gain, or mean number of Haemonchus contortus nematodes in the abomasum were observed. The results highlighted that carvacrol and limonene in the feed of lactating ewes effectively increased the weight gain of their twin suckling lambs, presumably due to the ewes’ energy status improvement. At the same time, further studies are needed to elucidate the effects of carvacrol and limonene against gastrointestinal parasites.

**Abstract:**

There is currently worldwide interest in phytogenic feed supplements (PFSs) because they can lead to improved animal production. The scope of the present study was to observe the impact of a feed supplement containing carvacrol (CAR) and limonene (LIM) on the performance and parasitological status of sheep. The feed supplement decreased the plasma levels of β-hydroxybutyrate (*p* < 0.001), triglycerides (*p* = 0.014), nonesterified fatty acids (*p* = 0.021), and fructosamine (*p* = 0.002) in lactating ewes after 42 days of supplementation, while the average live weight (*p* = 0.002) and average daily weight gain (*p* = 0.001) of their twin suckling lambs increased significantly by the end of the study. In another experiment, fattening lambs fed the same supplement showed a decrease in fecal egg number of gastrointestinal nematodes (*p* = 0.02) but no differences in live weight, average daily gain, or mean number of *Haemonchus contortus* nematodes in the abomasum. The results highlighted that the inclusion of carvacrol and limonene in the feed of lactating ewes effectively increased the weight gain of the suckling lambs, presumably due to the ewes’ improved energy, but further studies are needed to elucidate the effects of carvacrol and limonene against gastrointestinal parasites.

## 1. Introduction

The application of phytogenic products such as herbs, botanicals, essential oils, and oleoresins is an increasingly important aspect of animal nutrition. Phytogenic feed supplements (PFSs) contain plant secondary metabolites (PSMs), which are bioactive compounds with a wide range of physiological effects. Herbs can improve animal production as they can promote animal growth, boost the immune system, enhance reproduction, and improve energy efficiency from feed conversion [1,2]. The incorporation of plant bioactive compounds into an animal’s diet can change the composition of the gut microbiome. In the case of ruminants, the effects of PSMs on ruminal fermentation can include the modification of volatile fatty acid (VFA) concentration, methane emission, and ammonia–nitrogen concentration [3,4]. In vitro and in vivo studies have shown that pure or blended PSMs can manipulate rumen microbiota in advantageous and, in some cases, detrimental ways, with various effects on animal performance and production [4]. Moreover, their modes of action highly depend on the chemical structure and dose and the species to which they are applied [5].

PSMs have also been used since ancient times to prevent diseases and restore health. Both human and veterinary medicines use PSMs because of their antimicrobial, antiparasitic, and insecticidal effects [2,6,7]. The anthelmintic resistance (AR) of gastrointestinal nematodes (GINs) is responsible for some of the most significant economic losses and animal health problems in the small ruminant sector [8], and conventional drug use is becoming less effective for controlling GINs in most of the world. There are several factors that promote AR: the frequent application of the same drug, underdosing and lack of efficacy testing. For this reason, farmers need to apply integrated defensive methods to diminish their losses appreciably [9,10]. In these systems, the feeding of PFSs containing PSMs with anthelmintic (AH) properties could be an important tool [11].

The objective of our study was to evaluate the effects of an experimental non-commercially available phytogenic feed supplement (produced by ADEXGO Kft., Balatonfüred, Hungary) containing two terpenic components, carvacrol (CAR) and limonene (LIM), and Hydropalm (Norel S.A., Madrid, Spain) as a vehicle. The experimental design focused on animal performance and the parasitological status of sheep on a conventional Hungarian farm system, i.e., grouped animal feeding, grazing, and keeping. This pilot study was based on the idea that in real-world contexts, animal health problems (e.g., worm infection) and nutritional diseases (e.g., ketosis) often cluster because of common risk factors (e.g., inadequate management). Both parasite infections and malnutrition usually occur in Hungarian sheep herds during the lambing season and within the grazing lamb fattening system. The worm burden and malnutrition often interact in ways that make their combined practical and medical impacts much greater than the sum of their parts. This multimorbidity requires a complex parasitological and nutritional approach, which has also been applied to human disease map clustering [12]. Thus, we evaluated the effect of a PFS containing CAR and LIM on the performance and blood parameters of lactating ewes, and the anthelmintic properties of this PFS on fattening lambs with naturally acquired GIN infections.

## 2. Materials and Methods

Both studies were implemented in accordance with the guidelines of 32/1999 decree of the Hungarian Ministry of Agriculture, and the European Communities Council Directive (86/609 EEC). The experimental protocols were authorized by the Food Chain Safety and Animal Health Department of the Somogy County Government Office, under the permission number SO/31/000043-7/2023. A written information was given to both contributing farmers for the purpose of obtaining their informed consent.

### 2.1. Experiment 1

The aim of the first study was to observe the effect of the PFS on the blood parameters of lactating ewes and the average daily weight gain of their single and twin suckling lambs over the first 42 days of lactation.

#### 2.1.1. Animals, Feeding, and Husbandry System

The study was performed on a private Hungarian milking farm with Tzigai sheep. Thirty ewes were chosen to determine the effects of the PFS containing CAR and LIM. The study animals were separated from the rest of the breeding stock according to their expected lambing time and age (4–5 years old). The ewes were housed in two partitioned groups (CTR = control, *n* = 15; SUP = supplemented *n* = 15) in the same sheepfold with the same feeding and watering facilities and regime. The herd veterinarian admitted the animals to the study by the clinical signs of the upcoming lambing. The animals were randomly sorted into groups. The experimental period consisted of an adaptation phase to the feed supplement (ca. 4 days) before lambing followed by 42 days of supplemented feeding during lactation. For all ewes, the experimental feeding regime started before lambing and finished at weaning. The daily diet was grass hay fed ad libitum (approximately 2.5 kg daily intake) plus 1 kg of concentrate per ewe (Table 1) (producer: Agrofeed Ltd., Győr, Hungary). Detailed botanical and nutrient content of the hay were not determined. In the SUP group, all ewes received 0.5 g PFS per day mixed with the concentrate once a day between 8 and 9 a.m., while water and salt lick were provided ad libitum. This feeding regime modelled the general technology of the study farm. During the study, the animals were checked three times per day.

#### 2.1.2. Blood Sampling and Measurements

Blood serum analysis was conducted at the beginning (day 1) and at the end (day 42) of the experimental feeding period to determine the influence of the PFS on a sheep’s metabolic processes. Blood samples were collected within the framework of routine diagnostic monitoring, which was carried out by a licensed veterinarian. Blood samples were collected from the jugular vein of each ewe and left at room temperature for 60 min to coagulate. For blood collection, we used Vacuette^®^ CAT Serum Clot Activator 9 mL tube with 21G × 1″ blood specimen collection needle and blood collection needle holder (Greiner Bio-One International GmbH, Kermsmünster, Austria).

The serum samples were analyzed for aspartate aminotransferase (AST), alanine transaminase (ALT), alkaline phosphatase (ALP), gamma-glutamyl transferase (GGT), creatinine kinase (CK), lactate dehydrogenase (LDH), albumin (ALB), total protein (TP), β-hydroxybutyrate (BHB), total cholesterol (CHOL), triglycerides (TRIG), nonesterified fatty acids (NEFAs), glucose (GLU), fructosamine (FRA), and urea (UREA). The samples were analyzed in a private laboratory (“Vet-Med-Labor” Diagnostic Laboratory, Budapest, Hungary) using a Hitachi 912 Chemistry Analyser (Roche Diagnostic, Indianapolis, IN, USA).

#### 2.1.3. Live Weight and Weight Gain Determination

Indirect evaluation of the effect of the PFS was accomplished by measuring the weight of the suckling lambs at the beginning (day 1), after 3 weeks (day 21), and at the end (day 42) of the experimental feeding period. After lambing, the number of lambs (single or twins) was recorded. The average live weight (ALW) and average daily weight gain (ADWG) were determined separately for the two groups of lambs (single or twins). The 24 h body weight of the lambs was measured, and the mean was calculated at the beginning (ALW1) and on day 21 (ALW21) and day 42 (ALW42). The accuracy of the live weight measurement was ±0.1 kg. The ADWG values were calculated for the first part of the study period (days 1–21) and for the whole study period (days 1–42).

### 2.2. Experiment 2

The aim of the second study was to observe the effect of the PFS on the average daily weight gain and parasitological status of fattening lambs with naturally acquired GIN infections.

#### 2.2.1. Animals, Feeding, and Husbandry System

The same PFS was used as in the first assay to assess its effects on the parasitological status and weight gain of weaned lambs. A 32-day study was performed at the same sheep farm with 24 crossbred weaned lambs (Hungarian merino × Racka × Suffolk) aged 2.5 to 3 months. The animals had naturally acquired GIN infections and were stratified by live weight. Parasitological status was determined via fecal egg count (FEC). Lambs were housed in two partitioned groups (CTR = control, *n* = 12; SUP = all animals supplemented daily with 0.5 g PFS, *n* = 12) in the same sheepfold with the same feeding and watering facilities and regime. The proportion of females and males was equal in both groups (CTR and SUP). The animals were sorted into groups by live weight to assure a similar average baseline weight in each group. The average live weight of CTR and SUP lambs was 27.4 kg ± 1.97 and 26.8 ± 1.66, respectively.

The experimental period consisted of a 4-day adaptation phase before the experimental feeding regime began, followed by a 28-day fattening period. The daily diet was grass hay fed ad libitum (approximately 1 kg daily) and 1 kg/animal concentrated fattening lamb feed (Table 1). Detailed botanical and nutrient content of the hay was not determined. Water and salt lick were provided ad libitum. This feeding regime modelled the general technology of the study farm. During the study, the animals were checked three times a day.

#### 2.2.2. Live Weight and Weight Gain Determination

The fattening lambs were individually weighed at the beginning (day 1) and at the end (day 28) of the experimental fattening period with ±0.1 kg accuracy. The average weights (ALW1 and ALW28) and the average daily weight gain (ADWG) of the groups (CTR and SUP) were calculated. The ADWG was calculated by subtracting the initial body weight from the final body weight, divided by the number of days (28) between the initial weighing day and the last weighing day. 

#### 2.2.3. Parasitological Methods

Fecal samples were collected directly from the rectum of each animal on day 1 and day 28. Every sample was handled separately, and the FECs were determined in the laboratory immediately. For strongyle egg counting, we used the modified McMaster method described by Kassai [13]. We measured 3 g (±0.1 g) of homogenized fecal material and mixed it thoroughly with 42 mL of tap water. The suspension was filtrated by a mesh (pore size 100 μm), and of the total volume immediately, a 15 mL filtrate was transferred in a 15 mL plastic tube (Avantor Performance Materials Poland SA, Gliwice, Poland). After 3 min centrifugation (rpm = 1000), the supernatant was decanted, and the tube was filled up to 15 mL with a flotation fluid (saturated ZnSO_4_ solution, specific gravity = 1.37). The remained sediment was stirred with the solution and pipetted into both compartments of the McMaster counting chambers (Ghislandi and Co., Budapest, Hungary).

The sensitivity of this technique was 15 eggs. At the end of the trial (day 42), the animals were slaughtered for commercial purposes and their abomasa were collected to determine the level of worm burden. For the parasitological examination, the organs were separated from the forestomach and the small intestine.

The parasitological examination was performed as described by Kassai [13]. Isolated worms were placed in 96% alcohol for preservation until morphological examination. Identification was performed using a light microscope at 40× and 100× magnification, based on Lichtenfels et al. [14].

#### 2.2.4. Statistical Analysis

For the blood parameters, live weight, and ADWG, the means of the CTR and SUP groups were compared using a two-sample Student’s t-test for independent samples. The normal distribution of the data was evaluated using the Shapiro–Wilk test. If a given variable had a non-normal distribution, we normalized it via logarithmic transformation using R statistical software version i386 3.3.3. In all statistical comparisons, differences were declared at the *p* ≤ 0.05 level of significance. For the statistical analysis of FEC and abomasal worms, the mean intensity between the CTR and the SUP groups was compared. Calculations were carried out using Quantitative Parasitology 3.0 software [15] with a 95% confidence interval (CI 95%).

#### 2.2.5. Phytogenic Feed Supplement (PFS)

The plant-based feed supplement, which contained two terpenic PSMs (CAR and LIM), was covered with a hydrogenated lipid coat to avoid the rapid release of essential oil compounds in the rumen, which would be disadvantageous for rumen microbial fermentation [4]. The proportion of PFS mix and lipid coat was 1:1. The feed supplement was analyzed using gas chromatography–mass spectrometry (GC–MS) and high-performance liquid chromatography (HPLC) to determine the quality and quantity of the main components. The concentrations of CAR and LIM in the PFS were 97.6 mg/g and 36.0 mg/g, respectively.

## 3. Results

### 3.1. Experiment 1

#### 3.1.1. Blood Parameters

During the study, three animals were excluded from the CTR group and two animals from the SUP group for reasons of lamb (*n* = 1) and ewe death (*n* = 1) and nonpregnancy (*n* = 3). Thus, data were collected from 25 animals (CTR: *n* = 12; SUP: *n* = 13). On day 1, none of the variables differed significantly (*p* > 0.05). As a result, the creatinine level was not compared between the groups on day 42. At the end of the feeding period (day 42), significant differences were found between the groups in BHB (*p* < 0.001), TRIG (*p* = 0.014), NEFAs (*p* = 0.021), and FRA (*p* = 0.002) (Table 2).

#### 3.1.2. Weight Gain of the Lambs

In the CTR group, 16 lambs were weaned (female, *n* = 9; male, *n* = 7) and the proportion of single (*n* = 8) and twin lambs (*n* = 8) was equal. In the SUP group, nine female and eight male lambs were weaned, in which nine animals were born singly and eight as twins. Until halfway through the trial, the PFS did not exert any effect on the live weight of either the single or twin lambs (ALW21). At the end of the study (ALW42), however, there were significant differences between the two twin groups (*p* = 0.002), with heavier weights in the SUP group (21.13) than in the CTR group (15.43). There was no significant difference in weight between the groups of CTR and SUP for the single lambs (*p* = 0.071).

A very similar tendency was detected in the case of ADWG between days 1 and 42. For the CTR and SUP groups of single lambs, there were no significant differences (*p* = 0.076), while for twin lambs, the ADWG differed significantly between the two groups (*p* = 0.001). The ADWG in the SUP group was approximately 1.5-fold greater than that in the CTR group (Table 3).

### 3.2. Experiment 2

#### 3.2.1. Live Weight and Average Daily Gain

At the end of the feeding experiment with the 2.5- to 3-month-old fattening lambs, there was no significant difference in ALW or ADWG between the groups (Table 4).

#### 3.2.2. Fecal Egg Count

During the study, a decrease in worm egg counts as a function of time was observed in both lamb groups (Table 4). However, lambs in the SUP group, which were given feed supplemented with PFS, had significantly lower fecal egg counts on day 28 than the control group (*p* = 0.02).

#### 3.2.3. Abomasal Nematode Burden

After slaughter, only *Haemonchus contortus* specimens were isolated from the abomasa; no other nematodes were present. The mean intensity in the CTR group was 680 (CI 95% = 544–852), while in the SUP group, this variable was 50% higher at 1052 (CI 95% = 260–3304). Despite this considerable difference, the two groups did not differ significantly (*p* > 0.05).

## 4. Discussion

### 4.1. The Effect of PFS on the Blood Parameters of Lactating Ewes and the Average Daily Gain of Single and Twin Suckling Lambs

An analysis of the blood samples suggested that the energy requirements for milk production were not entirely supported by the feed consumed, and the ewes were using fat reserves. This catabolism was observable by the elevated levels of TRIG, BHB, and NEFAs. In early lactation, ewes have high energy and protein requirements. In this period, the voluntary feed intake is moderate; therefore, foraging is not sufficient to meet the high demands associated with optimal milk production. For this reason, in the first weeks of lactation, animals generally increase their use of body reserves (e.g., fat and muscle) even if they are fed ad libitum [16]. The loss of body fat can reach up to 7 kg between days 12 and 41 of lactation, while protein loss can be approximately 0.4 kg. This amount of fat is sufficient to produce approximately 50 kg of milk, while the protein is only adequate to produce 6 kg of milk [17]. As a consequence of the energy deficit associated with lactation, fat tissue mobilization begins. This process results in increasing levels of fatty acids (FAs) in the blood. Under optimal circumstances, these are channeled into the citric acid cycle after oxidative deterioration to acetyl units that increase the availability of energy at the cellular level. If the level of oxaloacetate is too low owing to the intensive gluconeogenesis required to produce milk sugars, the concentration of acetyl-coenzyme A increases in the liver; therefore, the level of ketone bodies (e.g., BHB) also increases [18,19,20].

Nevertheless, the inclusion of the PFS in the diet of lactating sheep modified the levels of blood metabolites connected with lipid metabolism. On day 42, the levels of TRIG, BHB, and NEFAs were significantly lower in the SUP group than in the CTR group. The lower levels of the abovementioned metabolites in the SUP group meant that ewes in this group displayed less fat tissue usage for milk production compared to the CTR ewes.

We hypothesize that the supplement had a positive effect on the energy efficiency of ruminal fermentation and therefore improved the energy status of the ewes because the results of the blood parameters indicated decreased body lipid catabolism in the SUP group. In the presence of PFSs, ruminal fermentation may be altered, resulting in changes to the ratio of volatile fatty acid and protein metabolism [4]. If the concentration of PSM present in a PFS in the rumen is moderate, i.e., between 50 and 500 mg/L, the PFS can increase the energy efficiency of fermentation by hampering methane synthesis, decreasing the loss of ammonia and increasing the concentration of propionate and butyrate compared to acetate [4]. As the energy efficiency of the fermentation process increased, the energy status of the animals receiving feed supplemented with the PFS improved; therefore, they used up less of their reserves, e.g., fat tissue via lipid metabolism, as detected by the lower blood levels of TRIG, BHB, and NEFAs.

This hypothesis was supported indirectly by the comparison with the ADWG of single and twin suckling lambs. Although there was no difference between the CTR and SUP groups in the case of the single lambs (*p* = 0.668 on days 1–21, and *p* = 0.076 on days 1–42), a difference in the ADWG of the twin lambs (*p* = 0.001 on days 1–42) was observed between the two groups. Although some in vitro studies showed differing results [21,22,23], we assumed that the increased ADWG of twin lambs was attributable to the ewes’ higher milk production rate as the ingested PFS improved their energy status via better conversion efficiency in the rumen. Presumably, the main effect of the applied PFS on rumen fermentation was a decreased ammonia and methane concentration and not an improvement in the VFA ratio because the main compounds in the PFS, CAR and LIM were reported to have a small or disadvantageous effect on VFA production [24]. Although some studies concluded that the PSMs can exert an inhibitory effects on ruminal bacteria [25,26], others detected the adaptation of bacteria to carvacrol and other PSMs [27,28]. In our study, we found that the total length of the feeding period was adequate for the adaptation of bacteria to the small amount of CAR and LIM released into the rumen from the supplement, in agreement with the observations of Igimi and Nishimura [29] and Michiels et al. [30,31].

A comparison of the CHOL levels almost showed an significant difference between the CTR and SUP groups (1.86 and 1.61, respectively, *p* = 0.059, Table 2). Nazifi et al. [32] and Antunović et al. [33] studied the influence of age and reproductive status on blood metabolites and concluded that the higher CHOL levels were the consequence of either the greater transport of lipoproteins or of energy deficit and lipolysis.

Overall, the results of our study suggested that administration of 0.5 g/animal/day of a PFS containing CAR and LIM had an advantageous effect on the metabolic processes of lactating ewes. The levels of metabolites related to lipid catabolism, such as TRIG, BHB, and NEFAs, significantly decreased in the blood of the ewes that ingested the PFS. The cause might be that the body lipid catabolism was attenuated in animals that ingested the PFS because of a better metabolic energy status. This assumption was also supported by that fact that the twin lambs sucking milk from ewes that had received feed supplemented with the PFS mix had a higher live weight and ADWG compared to the twin lambs of ewes receiving unsupplemented feed. In the case of ewes receiving the PFS, it is possible that a higher ratio of energy was available for milk synthesis, resulting in a higher milk yield and a higher ADWG of their twin lambs. The PFS seemed to influence the ADWG of twin lambs, but it did not have an effect on the ADWG of single lambs. This fact can support the hypothesis that the PFS improved the energy status of the lactating ewes, because a negative energy balance in lactation occurs more frequently for ewes with twins than with a single lamb.

### 4.2. The Effect of PFS on the Average Daily Gain and Parasitological Status of Fattening Lambs with Naturally Acquired GIN Infection

In the second study, inclusion of the PFS in the diet of fattening lambs did not result in a significant change in either the average body weight (ALW) or the ADWG of the groups although there was a tendency (*p* = 0.097) towards a higher ADWG in the group receiving feed supplemented with the PFS. Based on this trend, a higher dose of the PFS or longer time period of administration could be applied in further studies to investigate whether it can induce a significant increase in the ADWG of fattening lambs. In a study by Tadayon et al. [34], the inclusion of 110 and 220 g/kg DM dried orange peel with 18.9 g/kg tannic acid equivalent limonene in the diet of fattening lambs resulted in a significantly higher ADWG. The authors concluded that the reasons for the improved growth were an increase in feed intake, improved digestibility of nutrients, ruminal microbial nitrogen production and nitrogen retention. Based on our study it is worth determining whether an increased dose of PFS could improve the performance of fattening lambs, and which dose would be optimal for this aim.

A limitation of our study was that we did not determine the botanical and analytical content of the additionally fed grass hay, which could also have influenced the average performance of the lambs despite the small amount they consumed during the study period.

Regarding the parasitological status of the fattening lambs, the average number of excreted eggs in the feces decreased over the study period for both groups. The reason for this could be that, before the study, the fattening lambs were maintained on a feeding regime that was inadequate for the requirements of the animals in both quality and quantity, as it contained grass hay ad libitum and ground maize as a concentrate. We supposed that the feeding regime before the study (1) did not meet the animals’ physiological and production demands and (2) did not ensure the proper functioning of the animals’ immune systems. During the 28 day study, the feed of fattening lambs was fortified with concentrated lamb feed, which might have supported the immune system and resulted in a reduction in egg numbers. Though neither the age-related improvement in anthelmintic resistance can be excluded, a high-quality feeding regime can be an effective tool to support the animals’ defenses against GIN infection [10].

Moreover, the PFS containing CAR and LIM proved to be an effective tool against parasites as the number of fecal eggs was significantly lower in the group of animals that received the PFS than those in the controls. The direct anthelmintic effect of the investigated low dose of PFS needs further in vitro study to determine the minimal effective concentration against *H. contortus* or other abomasal nematodes.

However, when investigating abomasal nematode burden, we did not observe an AH effect of the PFS on the number of *Haemonchus contortus* nematodes found during a parasitological examination of the abomasa. We suggest that this phenomenon was due to the resilience of one specific lamb (ear tag 619) in the SUP group. Resilience means a genetically determined tolerance to maintain performance in the face of a parasitic burden [13]. The ADWG of this animal (271.43 g/day) exceeded the group average. Despite its higher performance, the observed abomasal nematode burden was 4080 specimens compared to the mean intensity (1052 specimen) for the rest of the group. Thus, this high worm number skewed the SUP mean intensity and caused the nonsignificant result. We did not remove this outlier because parasites never distribute normally between hosts; therefore, the presence of one or more outliers is not unique. Nevertheless, the apparent inefficiency of the PFS against worm burden highlighted a limitation of our study: the sample size, which did not allow multiple repetitions and avoidance of the pen effect. A larger sample size might have produced a more unequivocal result. Based on in vitro efficiency studies, an appropriate sample size can be determined to improve the study design.

Our parasitological results were similar to those reported for other studies which suggests that CAR and LIM can have an anthelmintic effect on GIN infection in sheep [35,36]. These molecules can affect the eggshell and cuticle of nematodes, which can cause chemical and structural damage to the cell membranes. We hypothesized that the permeability of the altered membranes was impaired, which might cause dysfunctions in cell homeostasis, metabolic processes, enzymatic inhibition, and fecundity and even the death of adult worms, as supported by the work of de Moraes et al. [37] and Andre et al. [35] on *Schistosoma mansoni* and *Haemonchus contortus*. Both authors observed considerable cuticle lesions caused by CAR and LIM. In addition, they suggested that these lesions affected the locomotion and the fecundity of the helminths. Further investigations are needed to determine the minimal effective concentrations of CAR and LIM against one of the most important abomasal nematodes, *H. contortus*. The results of these future studies might provide new tools for anthelmintic strategies.

## 5. Conclusions

Overall, we concluded that fortification of the feed of lactating ewes with a PFS containing CAR and LIM significantly improved weight gain in their suckling twin lambs. In lactating ewes, the PFS reduced the blood levels of metabolites associated with lipid catabolism; therefore, the effect of the PFS on the twin lambs’ performance was presumably due to the improved energy status of the lactating ewes. In the case of fattening lambs, there was a tendency for the PFS to improve the average weight gain of the animals; therefore, higher doses of the PFS or a more prolonged time of exposure should be included within the scope of the next study. Moreover, including the PFS in the feed of fattening lambs with naturally acquired GIN infections reduced the number of eggs counted in the feces. However, further studies are needed to elucidate the reasons why the effectiveness of CAR and LIM was restricted against the abomasal nematode *Haemonchus contortus*.

## Figures and Tables

**Table 1 vetsci-10-00369-t001:** Analysed nutrient content of the concentrate in Experiment 1 and Experiment 2 (as-fed basis).

	Experiment 1 ^1^	Experiment 2 ^2^
Dry matter, g/kg feed	894.2	892.5
Crude protein, g/kg	200.9	149.6
Crude fiber, g/kg	91.3	78.8
Crude fat, g/kg	92.8	36.6
Crude ash, g/kg	72.1	75.1
Ca, g/kg	11.6	12.8
P, g/kg	5.4	5.7
NE_m_, MJ/kg	6.71	7.11
NE_l_, MJ/kg	7.32	-
NE_g_, MJ/kg	7. 41	4.74

NEm = Net energy for maintenance; NEl = Net energy for lactation; NEg = Net energy for gain. ^1^ Ingredients: corn (36.0%), extracted rapeseed meal (18.1%), alfalfa meal (13.0%), sunflower meal (11.0%), wheat (10.0%), soybean meal (9.0%), CaCO_3_ (1.7%), premix (0.7%), NaCl (0.5%) (producer: Agrofeed Ltd., Hungary), ^2^ Ingredients: corn (30.0%), extracted rapeseed meal (18.2%), alfalfa meal (13.0%), sunflower meal (11.0%), soybean meal (10.0%), wheat (8.0%), hydrogenated fat (7%), CaCO_3_ (1.7%), premix (0.6%), NaCl (0.5%) (producer: Agrofeed Ltd., Hungary).

**Table 2 vetsci-10-00369-t002:** Examined blood parameters of the ewes on day 42.

Parameters ^1^	Reference Values	CTR (*n* = 12)	SUP (*n* = 13)	SEM ^2^	*p* Value
AST, IU/L	60–280	130.25	127.85	8.42	0.624
ALT, IU/L	26–34	15.92	13.46	0.659	0.093
ALP, IU/L	63–387	345.17	292.23	24.77	0.318
GGT, IU/L	20–52	57.67	61.15	2.73	0.559
CK, IU/L	100–547	307.08	313.85	26.11	0.462
LDH, IU/L	238–440	1831.67	1572.46	65.2	0.103
ALB, g/L	2.4–3	30.09	28.96	0.454	0.191
TP, g/L	60–79	87.68	88.16	1.568	0.624
BHB, mmol/L	0.45–0.64	0.93	0.48	0.055	<0.001
TRIG, mmol/L	0.06–0.34	0.17	0.11	0.01	0.014
NEFAs, mmol/L	0.1–0.5	0.21	0.11	0.025	0.021
CHOL, mmol/L	1.35–1.97	1.86	1.61	0.061	0.059
GLU, mmol/L	2.78–4.44	1.59	1.32	0.081	0.087
FRA, μmol/L	1250–1360	158.5	145.62	2.087	0.002
UREA, mmol/L	2.8–7.1	7.17	7.74	0.217	0.245

^1^ AST = aspartate aminotransferase, ALT = alanine transaminase, ALP = alkaline phosphatase, GGT = gamma glutamyl transferase, CK = creatinine kinase, LDH = lactate dehydrogenase, ALB = albumin, TP = total protein, BHB = beta-hydroxybutyrate, TRIG = triglycerides, NEFAs = nonesterified fatty acids, CDHOL = total cholesterol, GLU = glucose, FRA = fructosamine, UREA = urea. ^2^ SEM = standard error of the mean.

**Table 3 vetsci-10-00369-t003:** Average live weight and average daily gain of single and twin lambs in Experiment 1.

Parameters ^1^	Weight of Single Lambs (kg)				Weight of Twin Lambs (kg)			
	CTR	SUP	SEM ^2^	*p* Value	CTR	SUP	SEM	*p* Value
ALW1, kg	3.86	3.7	0.17	0.439	3.56	3.46	0.11	0.723
ALW21, kg	10.65	10.18	0.275	0.447	8.05	9.15	0.415	0.542
ALW42, kg	19.89	21.45	0.393	0.071	15.43	21.13	0.898	0.002
ADWG (1–21 days), g/day	323.21	308.47	0.015	0.668	235.71	270.83	0.019	0.47
ADWG (1–42 days), g/day	381.55	422.75	0.011	0.076	283.94	420.54	0.162	0.001

^1^ ALW1 = average live weight on day 1; ALW21 = average live weight on day 21; ALW42 = average live weight on day 42; ADWG = average daily weight gain, ^2^ SEM = standard error of the mean.

**Table 4 vetsci-10-00369-t004:** Live weight and fecal egg count of fattening lambs in Experiment 2.

Parameters ^1^	CTR	SUP	SEM ^2^	*p* Value
ALW1, kg	27.1	26.8	0.378	0.423
ALW28, kg	32.8	33.2	0.495	0.659
ADWG (1–28 days), g/day	191.1	229.5	0.011	0.097
Fecal egg count on day 1	2058	1549	416.4	0.555
Fecal egg count on day 28	1190	480	155.2	0.02

^1^ ALW1 = live weight on day 1; ALW28 = live weight on day 21; ADWG = average daily weight gain. ^2^ SEM = standard error of the mean.

## Data Availability

The data presented in this study are available on request from the corresponding author.

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
