# Peer review of "Evaluation of a Phytogenic Feed Supplement Containing Carvacrol and Limonene on Sheep Performance and Parasitological Status on a Hungarian Milking Sheep Farm"

_vetsci, 2023, doi:10.3390/vetsci10060369_

Round 1
Reviewer 1 Report
The research is simple and with great practical application.
The language is accessible and scientifically sustainable.
Similar results have already been described with other animal models, such as chickens, layers and swine.
The results of this research do not have the potential to change the " state-of-the-art", but consolidate a recommendation for the use of essential oils in diet and animals.
Author Response
Thank you very much for the supportive review.
Reviewer 2 Report
This paper describes two experiments in which phytogenic feed supplements were provided to sheep. In the first experiment, the effect of supplementation on blood parameters and weight measurements was explored. In the second experiment the effect of supplementation on weight measurements and parasitological parameters was tested. This is a well written paper, with interesting results that warrant publication. Below are some minor comments and recommendations that authors may like to consider.
Line 49/50 – Conventional drug use is still somewhat effective in many locations. Maybe rephrase to something like “The use of anthelmintic drugs has resulted in the development of anthelmintic resistance, reducing the efficacy of conventional treatments in most parts of the world.
Line 50/51 – Several factors that ARE promoting AR
Line 57 – Change to: (Not commercially available)
Line 56-61 – This is a long sentence that is a little hard to follow. Consider breaking it up.
Line 94 – How was the 0.5g of PFS delivered to each sheep? Individually? As part of the concentrate?
Line 158 – 169: I am not familiar with the Kassai methods. Please add a very brief description of these methods.
Line 323-326 – Rephrase: You don’t know for sure that providing more of the supplement will result in a significant increase in ADWG. Your results do suggest that this hypothesis is worth testing though.
Line 326 – How much LIM did Tadayon provide their sheep? That information could support the previous sentence.
Line 342-344: Do you know for sure that the supplement improved the sheep’s immune systems? Maybe it had a direct effect on the worms, reducing their capacity to produce eggs?
Author Response
Line 49/50 – Conventional drug use is still somewhat effective in many locations. Maybe rephrase to something like “The use of anthelmintic drugs has resulted in the development of anthelmintic resistance, reducing the efficacy of conventional treatments in most parts of the world.
The suggestion is accepted. We corrected the indicated sentence.
Line 50/51 – Several factors that ARE promoting AR
The suggestion is accepted. We corrected the indicated sentence.
Line 57 – Change to: (Not commercially available)
The suggestion is accepted. We corrected the indicated sentence.
Line 56-61 – This is a long sentence that is a little hard to follow. Consider breaking it up.
The suggestion is accepted. We corrected the indicated sentence.
Line 94 – How was the 0.5g of PFS delivered to each sheep? Individually? As part of the concentrate?
The suggestion is accepted. We gave the missing information.
Line 158 – 169: I am not familiar with the Kassai methods. Please add a very brief description of these methods.
The suggestion is accepted. We added the method’s description.
Line 323-326 – Rephrase: You don’t know for sure that providing more of the supplement will result in a significant increase in ADWG. Your results do suggest that this hypothesis is worth testing though.
The suggestion is accepted. We corrected the indicated sentence.
Line 326 – How much LIM did Tadayon provide their sheep? That information could support the previous sentence.
The suggestion is accepted. We gave the missing information.
Line 342-344: Do you know for sure that the supplement improved the sheep’s immune systems? Maybe it had a direct effect on the worms, reducing their capacity to produce eggs?
The suggestion is accepted. We gave some additional informations in this context.
Reviewer 3 Report
The abstract need additional detail re the timescale of the lamb response in the ewe study - it currently suggests this is throughout but the growth responses were only in the latter half of this study period.
l57 should this read (not commercially available) rather than (commercially no available).
l58 you mention that the supplement also contains hydropalm (trade name? please include generic description)but then ignore this for the rest of the paper and only discuss the role of CAR and LIM in the supplement.
l89-l92 this methodology appears to contradict itself. It states there was a period of adaptation to the feed supplement for 4 days before lambing but then states that supplementation started at lambing. Was this an acclimitisation to the experimental pens, or the basal diet? The other difficulty with starting a set days prelambing is the prediction and variability of lambing dates, this would make the acllimitisation period variable and it would best be to quote the mean +/- sd or median and interquartile range for each group. How were the sheep allocated to which group are we random or was this stratified by the time on acclimitisation period?
l94 Is this supplement dose correct (0.5g) and how was this presented to the sheep, was it mixed into feed or direct to animal delivered?
Table 1 has the concentrate composition but what was the composition of the hay? and the treatment. Would be better presented as a complete diet. Was hay intake measured? Were any intakes different between groups - if intakes not measured then need to tone down metabolic difference interpretation and effect could be from an increased feed intake
l112 tube type and supplier etc
l114-120 We have analyser details but need supplier details for individual kits for each assay or if this is not known need to reference the specific laboratory used so that SOPs etc can be followed up.
l142 how were the lambs allocated to groups? If stratified according to weight then show group weight parameters.
l144 the basal diet needs to be characterised and presented and given the later comments it would also be good to show the pre-experimental diet. Were lambs pre experiment just fed maize or was there a forage component?
l171 You have 1 pen of each treatment and then are carrying out tests for independent samples? Are the individual animals in this set up truely independent as pen effects can confound the true treatment effects. Whilst not necessarily an issue in itself this does need to be discussed as a limitation of this study. Ideally we needed individual feeding or at least replicate pens of each treatment. The pen is strictly your experimental unit unless all animals housed together in 1 pen and the treatment delivered in a direct to animal manner.
l180 and elsewhere I am slightly confused by PFS, PMS and your treatment these are used as generic terms in discussion and intro but also seem to be terms indicating your treatment, this needs some clarification.
Table 2, you are using non-standard abbreviations for your metabolites I would suspect AST, ALT, ALP, GGT, CK, LDH, ALB, TP, BHB(or BOHB), TRIG, NEFA, CHOL, GLU, FRUC, UREA. It would be good to include normal ranges for these paprameters here.
Normal ranges should be included in the paper within the text if not in table (or both). You are providing some of this interpretation but not your normal ranges and the sources that you are using, making reader evaluation of your interpretation difficult.
You have low group numbers and it maybe useful to carry out power calculations to show the power of your analysis and/or the ideal group numbers for this response.
Tables 3/4 you need to include the group numbers (n=?)
The discussion is a little vague and non-specific in places eg l291 'disadvantageous effect on VFA production' gives no detail what is the issue is it a change in VFA profile? a decrease in VFA production? smacks of finding a line in a paper rather than understanding a potential cause. How does the supplemnt have a positive effect on the nergy efficency og ruminal fermentation (l265) - no detail given. PFS, PSM are confused in discussion.
re acclimitisation l295 4 days is poss to short for major dietary change which may effect the lack of initial response?
l335 'only ground maize' - no forage component?
In grazing situations there will be a background parasitic load. There could be a load effect rather than just a immunity response - seems an over simple approach.
l348-354 - prob worth doing the stats excluding this animal and then mentioning here if removing the outlier does have an effect on the statistical output?
l388 How did you get your sheep (experimental subjects) to give informed consent to this work?
Author Response
The abstract need additional detail re the timescale of the lamb response in the ewe study - it currently suggests this is throughout but the growth responses were only in the latter half of this study period.
The suggestion is accepted. We completed the abstract with the required information.
l57 should this read (not commercially available) rather than (commercially no available).
The suggestion is accepted. We corrected the indicated sentence.
l58 you mention that the supplement also contains hydropalm (trade name? please include generic description)but then ignore this for the rest of the paper and only discuss the role of CAR and LIM in the supplement.
The suggestion is accepted. We gave the missing information.
l89-l92 this methodology appears to contradict itself. It states there was a period of adaptation to the feed supplement for 4 days before lambing but then states that supplementation started at lambing. Was this an acclimitisation to the experimental pens, or the basal diet? The other difficulty with starting a set days prelambing is the prediction and variability of lambing dates, this would make the acllimitisation period variable and it would best be to quote the mean +/- sd or median and interquartile range for each group. How were the sheep allocated to which group are we random or was this stratified by the time on acclimitisation period?
The suggestion is accepted. We gave the missing information.
l94 Is this supplement dose correct (0.5g) and how was this presented to the sheep, was it mixed into feed or direct to animal delivered?
The suggestion is accepted. We gave the missing information.
Table 1 has the concentrate composition but what was the composition of the hay? and the treatment. Would be better presented as a complete diet. Was hay intake measured? Were any intakes different between groups - if intakes not measured then need to tone down metabolic difference interpretation and effect could be from an increased feed intake
The suggestion is accepted. We emphasised the limitations of the study.
l112 tube type and supplier etc
The suggestion is accepted. We gave the missing information.
l114-120 We have analyser details but need supplier details for individual kits for each assay or if this is not known need to reference the specific laboratory used so that SOPs etc can be followed up.
The suggestion is accepted. We gave the missing information.
l142 how were the lambs allocated to groups? If stratified according to weight then show group weight parameters.
The suggestion is accepted. We gave the missing information.
l144 the basal diet needs to be characterised and presented and given the later comments it would also be good to show the pre-experimental diet. Were lambs pre experiment just fed maize or was there a forage component?
The suggestion is accepted. We corrected the indicated sentence.
l171 You have 1 pen of each treatment and then are carrying out tests for independent samples? Are the individual animals in this set up truely independent as pen effects can confound the true treatment effects. Whilst not necessarily an issue in itself this does need to be discussed as a limitation of this study. Ideally we needed individual feeding or at least replicate pens of each treatment. The pen is strictly your experimental unit unless all animals housed together in 1 pen and the treatment delivered in a direct to animal manner.
The suggestion is accepted. We emphasised the limitations of the study.
l180 and elsewhere I am slightly confused by PFS, PMS and your treatment these are used as generic terms in discussion and intro but also seem to be terms indicating your treatment, this needs some clarification.
The suggestion is accepted. We clarified the indicated acronyms.
Table 2, you are using non-standard abbreviations for your metabolites I would suspect AST, ALT, ALP, GGT, CK, LDH, ALB, TP, BHB(or BOHB), TRIG, NEFA, CHOL, GLU, FRUC, UREA. It would be good to include normal ranges for these paprameters here.
The suggestion is accepted. We gave the missing information.
Normal ranges should be included in the paper within the text if not in table (or both). You are providing some of this interpretation but not your normal ranges and the sources that you are using, making reader evaluation of your interpretation difficult.
See as above.
You have low group numbers and it maybe useful to carry out power calculations to show the power of your analysis and/or the ideal group numbers for this response.
The note is accepted. We emphasised the limitations of the study.
Tables 3/4 you need to include the group numbers (n=?)
The suggestion is accepted. We emphasised the limitations of the study.
The discussion is a little vague and non-specific in places eg l291 'disadvantageous effect on VFA production' gives no detail what is the issue is it a change in VFA profile? a decrease in VFA production? smacks of finding a line in a paper rather than understanding a potential cause. How does the supplemnt have a positive effect on the nergy efficency og ruminal fermentation (l265) - no detail given. PFS, PSM are confused in discussion.
The note is accepted. The discussion is completed with additional information and emphasised the limitations of the study.
re acclimitisation l295 4 days is poss to short for major dietary change which may effect the lack of initial response?
The note is accepted. The discussion is completed with additional information.
l335 'only ground maize' - no forage component?
The suggestion is accepted. We gave the missing information.
In grazing situations there will be a background parasitic load. There could be a load effect rather than just a immunity response - seems an over simple approach.
The suggestion is accepted. We gave the missing information.
l348-354 - prob worth doing the stats excluding this animal and then mentioning here if removing the outlier does have an effect on the statistical output?
The note is accepted. The discussion is completed with additional information.
l388 How did you get your sheep (experimental subjects) to give informed consent to this work?
The suggestion is accepted. The manuscript is completed with additional information.
Round 2
Reviewer 3 Report
The authors appear to have addressed all points satisfactorially.